Effects of chronic prazosin, an alpha-1 adrenergic antagonist, on anxiety-like behavior and cortisol levels in a chronic unpredictable stress model in zebrafish (Danio rerio)

O’Daniel Michael P.
Petrunich-Rutherford Maureen L. mlpetrun@iun.edu
Department of Psychology, Indiana University Northwest , Gary , IN , United States of America
Connor Mark
Electronic publication date: 2020 Jan 31
Publication date: 2020
Volume: 8
Electronic Location ID: e8472
Received 2019 Sep 10; Accepted 2019 Dec 27
Copyright: ©2020 O’Daniel and Petrunich-Rutherford
Copyright year: 2020
Copyright holder: O’Daniel and Petrunich-Rutherford
License: This is an open access article distributed under the terms of the Creative Commons Attribution License, which permits unrestricted use, distribution, reproduction and adaptation in any medium and for any purpose provided that it is properly attributed. For attribution, the original author(s), title, publication source (PeerJ) and either DOI or URL of the article must be cited.
License URL: https://creativecommons.org/licenses/by/4.0/

Keywords: Anxiety, Stress, Zebrafish, Cortisol, Anxiety-like behavior, Chronic unpredictable stress, Prazosin, Alpha-1 adrenergic, Norepinephrine, Animal model

Funding: IU Northwest Undergraduate Research Fund Faculty Grant-in-aid of Research Summer Faculty Fellowship This work was supported by the IU Northwest Undergraduate Research Fund, Faculty Grant-in-aid of Research, and Summer Faculty Fellowship. The funders had no role in study design, data collection and analysis, decision to publish, or preparation of the manuscript.

==============================
Post-traumatic stress disorder (PTSD) is often associated with significant neuroendocrine dysfunction and a variety of other symptoms. Today, there are limited efficacious treatment options for PTSD, none of which directly target the dysfunction observed with the hypothalamic-pituitary-adrenal (HPA) axis. The development of new pharmacological treatments is expensive and time consuming; thus, there is utility in repurposing compounds already approved for use in other conditions. One medication in particular that has shown promise for the alleviation of PTSD symptoms is prazosin, an alpha-1 adrenergic receptor antagonist used to treat hypertension. While there have been many studies indicating the efficacy of prazosin in the treatment of PTSD symptoms, no studies fully elucidate mechanisms elicited by this treatment, nor is it clear if prazosin normalizes neuroendocrine dysfunction associated with trauma exposure. The use of zebrafish (Danio rerio) has been growing in popularity, in part, due to the homology of the stress response system with mammals. In this study, the zebrafish model was utilized to determine behavioral and biological changes induced by chronic unpredictable stress (CUS) and how these effects could be modulated by chronic prazosin treatment. The results indicated that 7d of CUS increased anxiety-like behavior in the novel tank test and decreased basal levels of cortisol. Chronic (7d) prazosin treatment decreased anxiety-like behaviors overall but did not appear to affect CUS-induced changes in behavior and basal cortisol levels. This suggests that the clinical effectiveness of prazosin may not normalize dysregulated stress responses prevalent in many patients with PTSD, but that prazosin-induced relief from anxiety in stress-related conditions may involve an alternative mechanism other than by normalizing neuroendocrine dysfunction.

Introduction

Post-traumatic stress disorder (PTSD) is a disorder that inhibits day-to-day functionality due to a plethora of disrupting symptoms including flashbacks, vivid nightmares, mood alterations, and hypervigilance (Bisson et al., 2015). According to the World Health Organization (WHO) World Mental Health Surveys, the overall lifetime prevalence of PTSD is estimated to be around 3.9%, increasing to about 5.6% of trauma-exposed individuals (Koenen et al., 2017). The most common traumas associated with PTSD are interpersonal traumas, including rape and other types of sexual assault (Kessler et al., 2017). Current evidence-based treatment plans for PTSD include cognitive behavioral therapy and pharmacological options, mainly serotonin reuptake inhibitors (Lancaster et al., 2016); however, many cases are resilient to or respond ineffectively to first-line treatments (Foa et al., 2009). It appears that there is a growing trend of prescribing benzodiazepines for the management of PTSD symptoms, at least in U.S. active duty service members (Loeffler et al., 2018); however, this practice is associated with concerning outcomes, such as increased suicide risk (Deka et al., 2018). In addition, it is well known that chronic use of benzodiazepines is associated with physiological dependence and subsequent withdrawal symptoms upon treatment discontinuation (Pétursson, 1994).

Due to the resilience to treatment and concerning growth in use of sedative prescription medications, the need for novel treatments for PTSD has become apparent. Because of the exorbitant cost of effective novel drug synthesis and testing, there has been growing interest in the repurposing of compounds already approved by the Food and Drug Administration in the treatment of other conditions (Papapetropoulos & Szabo, 2018). Prazosin, an alpha-1 adrenergic receptor antagonist originally utilized in the treatment of hypertension, has been shown to alleviate clinical symptoms of PTSD (Ahmadpanah et al., 2014; De Berardis et al., 2015; Green, 2014; Koola, Varghese & Fawcett, 2014; Simon & Rousseau, 2017; Singh et al., 2016; Writer, Meyer & Schillerstrom, 2014). However, there have been few investigations into the mechanisms behind the clinical efficacy of this compound, particularly involving physiological symptoms associated with chronic stress exposure.

The physiological stress response in mammals is largely controlled by the hypothalamic-pituitary-adrenal (HPA) axis. The HPA axis acts in a negative feedback loop, wherein the hypothalamus signals the pituitary gland via corticotropin-releasing hormone (CRH) to stimulate the release of adrenocorticotropic hormone (ACTH) into the bloodstream. ACTH then acts peripherally to stimulate the release of glucocorticoids (e.g., cortisol, corticosterone) from the adrenal glands to mobilize the body’s resources to deal with a stressor. Then, cortisol binds to glucocorticoid receptors in the hypothalamus, pituitary gland, and other upstream brain structures to attenuate the stress response. Studies indicate that the HPA axis is dysregulated in patients with PTSD (Mason et al., 1986; Pervanidou & Chrousos, 2010; Wichmann et al., 2017; Yehuda et al., 1990). There is evidence that prazosin helps to normalize HPA dysfunction in subjects in the early stages of alcohol withdrawal (Fox et al., 2012); however, it is unknown if prazosin would similarly alleviate HPA dysfunction associated with chronic stress exposure. Furthermore, the exact direction of chronic stress-induced dysregulation (i.e., upregulation or downregulation) is complex and likely dependent on a number of factors, such as biological sex and early life stress exposure (Dunlop & Wong, 2019). Thus, a more complete understanding of individual factors affecting stress-induced alterations in HPA functioning can be examined with animal models.

The stress response is heavily conserved amongst vertebrates. The zebrafish (Danio rerio) has been asserted as a viable model for stress related research because of the similarities in the physiological stress response (Clark, Boczek & Ekker, 2011). The hypothalamic-pituitary-interrenal (HPI) axis is considered to be the zebrafish analogue to the mammalian HPA axis (Nesan & Vijayan, 2013; Wendelaar Bonga, 1997). In addition, zebrafish have been growing in popularity for translational research due to genetic and physiological similarities to mammals in stress and anxiety-like behavioral responses (Caramillo et al., 2015). Zebrafish exposed to chronic unpredictable stress (CUS) may facilitate a better understanding of factors that may convey vulnerability to behavioral disorders that affect humans, such as major depressive disorder (Fulcher et al., 2017) and PTSD (Caramillo et al., 2015; Stewart et al., 2014). The stressors that are utilized in the CUS zebrafish model vary in intensity, duration of stress, and type of stress and include mechanical, chemical, and temperature changes. The stressors are randomized and are administered at different times. Several studies have indicated that CUS modeling in zebrafish elicits anxiety-like behaviors in a variety of testing paradigms (Chakravarty et al., 2013; Fulcher et al., 2017; Marcon et al., 2016; Marcon et al., 2018; Piato et al., 2011; Song et al., 2018). CUS also increases whole-body cortisol levels (Manuel et al., 2014; Marcon et al., 2016; Marcon et al., 2018; Piato et al., 2011; Song et al., 2018), although in one report, CUS was shown to increase basal cortisol levels only in male fish but did not significantly alter basal cortisol levels in female fish (Rambo et al., 2017). Thus, the CUS model in zebrafish could be used to examine the efficacy of repurposed compounds and clarify the mechanisms by which these compounds could alleviate physiological dysfunction associated with stress-related conditions.

The current study utilized the seven-day chronic unpredictable stress (CUS) model as reported in the literature (Manuel et al., 2014; Marcon et al., 2016; Marcon et al., 2018; Piato et al., 2011; Rambo et al., 2017) to replicate the effects of the protocol on anxiety-like behavior and basal cortisol levels. Then, in another experiment, chronic treatment with either prazosin or vehicle followed a week of CUS to examine whether prazosin would reverse any CUS-induced changes in hormones or behavior. It was hypothesized that CUS would increase basal levels of cortisol and elicit increases in anxiety-like behavior, as evidenced by increased freezing and decreased exploratory behavior in the novel tank test. It was also expected that chronic prazosin treatment would normalize both neuroendocrine and behavioral alterations observed after CUS exposure. The results from the current study could provide evidence for the mechanism of prazosin’s clinical efficacy and suggest that it may be a viable treatment option for individuals with HPA dysfunction associated with stress-related conditions, such as PTSD.

Methods and Materials

Animals and housing

Wild-type, adult, mixed-sex zebrafish (total N = 122) were purchased from Carolina Biological Supply (Burlington, NC). Upon delivery, zebrafish were randomly placed into housing tanks and allowed to acclimate to the facility for at least one week before any experimental procedures were initiated (Dhanasiri, Fernandes & Kiron, 2013). Zebrafish were housed in a stocking density of 5–7 fish per liter in 1.8L tanks and maintained in a two-shelf, stand-alone zebrafish housing rack purchased from Aquaneering (San Diego, CA). Fish were maintained on a 14 h:10 h light:dark cycle, with water kept at 27 ± 1 °C and pH of approximately 7.2. Other water quality parameters were measured biweekly, such as ammonia, nitrates, nitrites, alkalinity, and hardness, and were kept constant throughout the experiments. Fish were fed once per day with flake food and once per day with dried shrimp ground to a powder with a mortar and pestle. Feeding commenced around 9 a.m. each day, before any stress or drug procedures were conducted, except for on days of data collection. The fish were not fed prior to the behavioral testing. The total food weight given per day per fish approximated 4% of the average fish body weight. All procedures were carried out by following established recommendations (Harper & Lawrence, 2011; National Research Council, 2011; Westerfield, 2000).

Drugs and materials

Prazosin hydrochloride was manufactured by TCI America and purchased from VWR International (Radnor, PA). N-N-dimethylacetamide was manufactured by Frontier Scientific and purchased from Fisher Scientific (Hampton, NH).

Experiment 1: Chronic unpredictable stress on anxiety-like behavior and basal cortisol levels

Upon arrival to the facility, fifty zebrafish were randomly allocated into four separate 1.8L housing tanks. After at least 7 days of acclimation to the facility, two tanks of fish were randomly selected and subsequently exposed to the chronic unpredictable stress model for seven days. The other two tanks of fish served as untreated, unstressed controls. On the day after the completion of the chronic stress paradigm, fish from both control and stressed groups were placed in the novel tank test one at a time to assess anxiety-like behavior. Fish were immediately euthanized after the behavioral assessment and decapitated to assess basal levels of trunk cortisol. At the end of the experiment, N = 25 were exposed to chronic unpredictable stress and N = 25 were untreated for 7 days.

Experiment 2: Chronic unpredictable stress and chronic prazosin treatment on anxiety-like behavior and basal cortisol levels

Upon arrival to the facility, seventy-two zebrafish were randomly allocated into eight separate 1.8L housing tanks. After at least 7 days of acclimation to the facility, four tanks of fish were randomly selected and subsequently exposed to the chronic unpredictable stress model for seven days and the other four tanks of fish were not stressed (left unhandled) for seven days. Then, two tanks of the stressed fish and two tanks of the non-stressed fish were treated with prazosin for 30 min per day for the seven days following the CUS treatment. The other four tanks were exposed to vehicle treatment for 30 min per day for seven days. On the day after the completion of the drug or vehicle treatment, fish from all groups were placed in the novel tank test one at a time to assess anxiety-like behavior. Fish were immediately euthanized after the behavioral assessment and decapitated to assess basal levels of trunk cortisol. A total of three fish died during the course of the procedures (N = 2 from the unstressed/vehicle-treated group and N = 1 from the stressed/vehicle-treated group). At the end of the experiment, N = 18 were exposed to 7d chronic unpredictable stress followed by chronic (7d) prazosin, N = 17 were exposed to 7d chronic unpredictable stress followed by 7d vehicle, N = 18 were untreated for 7 days and then exposed to chronic (7d) prazosin, and N = 16 were untreated for 7 days and then exposed to vehicle for 7 days.

Chronic unpredictable stress (CUS) model

The chronic unpredictable stress (CUS) model was adapted from previously published procedures (Marcon et al., 2016; Marcon et al., 2018; Piato et al., 2011; Rambo et al., 2017). Seven different types of stressors were ordered at random. Fish in the stressed group were exposed to two stressors per day for seven days at random times between 9 a.m. and 4 p.m. (see Table 1 for schedule). The stressors included (1) tank changes (three times) in rapid succession, (2) cooling home tank water abruptly to 23 °C and maintaining that temperature with chilled system water for 30 min before placing tank back on the system, (3) heating home tank water abruptly to 33 °C and maintaining that temperature with heated system water for 30 min before placing tank back on the system, (4) lowered water (1 cm depth for 15 min), (5) net chasing in home tank (8 min chase, 15 min rest, 8 min chase), (6) crowding all fish from one home tank in 200 ml system water (9-13 fish total, density of 45–65 fish per liter) in a 250 ml beaker for 60 min, and (7) social isolation (individual fish were placed in 200 ml system water in 250 ml beakers separated by opaque dividers). Control fish were not stressed (not handled) and were not removed from the system for 7 days.

Table 1 Chronic unpredictable stress schedule for Experiment 1 and Experiment 2.

Seven stressors were randomized; fish in the stressed group were exposed to the cycle of stressors twice over 7 days. Subjects were exposed to stressors twice a day at random times during the light period between the times of 9 a.m. and 4 p.m.

Experiment Day	Stressor	Time of day(Experiment 1)	Time of day(Experiment 2)	
1	Tank changes (3 times)	11 a.m.	12 p.m.	
Cooling (23 °C, 30 min)	1 p.m.	1 p.m.	
2	Lowered water (15 min)	3 p.m.	2 p.m.	
Net chase (8 min + 15 min rest + 8 min)	4 p.m.	3 p.m.	
3	Crowding (250 ml beaker, 60 min)	10 a.m.	10 a.m.	
Heating (33 °C, 30 min)	2 p.m.	4 p.m.	
4	Social isolation (250 ml individual beakers, 45 min)	11 a.m.	9 a.m.	
Tank changes (3 times)	12 p.m.	11 a.m.	
5	Cooling (23 °C, 30 min)	3 p.m.	11 a.m.	
Lowered water (15 min)	4 p.m.	4 p.m.	
6	Net chase (8 min + 15 min rest + 8 min)	1 p.m.	11 a.m.	
Crowding (250 ml beaker, 60 min)	2 p.m.	4 p.m.	
7	Heating (33 °C, 30 min)	10 a.m.	10 a.m.	
Social isolation (250 ml individual beakers, 45 min)	12 p.m.	1 p.m.	

Drug treatment

Fish undergoing chronic administration of prazosin were gently netted from the home tank and individually placed into a 100 mL beaker containing 2 mg prazosin dissolved in 100 microliters of N,N-dimethylacetamide and 50 mL of system water for 30 min (Singh et al., 2013). This concentration was chosen due to the observed anxiolytic behavior in the light-dark test displayed by the fish acutely exposed to prazosin (Singh et al., 2013). In the current experiment, the drug exposure was repeated once daily for seven days; fresh drug solution was prepared for each fish for each exposure from a concentrated prazosin solution prepared at the beginning of the experiment. Subjects in the control group were subjected to similar handling and conditions although only exposed to the vehicle (100 microliters of N,N-dimethylacetamide in 50 mL system water in a 100 mL beaker for 30 min per day). Fresh vehicle solution was prepared each day. All fish were returned to their respective home tanks after the treatment and placed back on the system between daily treatment sessions.

Novel tank test (NTT)

On the day of the experiment (the day after the last episode of chronic stress for fish in Experiment 1 and the day after the last drug treatment for fish in Experiment 2), home tanks were removed from the system and moved into the experiment room adjacent to the housing facility. To minimize the impact of the stress of moving the tanks, fish were left to acclimate for at least 30 min before assessing behavior. The experimental room had the same lighting and temperature conditions as the housing room. Fish were individually netted and placed into a trapezoidal novel tank, the same size and dimensions as the home tanks (15.2 cm height × 27.9 cm top × 22.5 cm bottom × 7.1 cm width), for six minutes. The novel tank was filled with water from the system and was changed on each new day of data collection. The behavior of each fish was recorded and subsequently analyzed with BehaviorCloud motion-tracking software (Alia & Petrunich-Rutherford, 2019; Aponte & Petrunich-Rutherford, 2019; Pilehvar, Town & Blust, 2020). Total distance traveled (cm) and mean ambulatory speed (cm/s) were measured as markers of general motor activity; immobility duration (sec), the number of entries to the top of tank, time spent in top (sec), and distance traveled in the top (cm) and were used as markers of anxiety-like behavior (Cachat et al., 2010; Egan et al., 2009; Wong et al., 2010). The top of the tank was defined as the top 50% (approximately 7 cm) of the water column (total approximately 14 cm). Behavioral data collection and euthanasia of the subjects occurred between 9:30 a.m. and 2:30 p.m.

Euthanasia

Immediately after the novel tank test, fish were netted from the novel tank and placed individually in a 50 mL beaker with approximately 30 mL 0.1% (100 mg/L) clove oil in system water. Death occurred within 5 to 10 s of introduction to the solution and was determined upon visual examination for cessation of opercular (gill) movement and nonresponse to tactile stimulation (Davis et al., 2015). The fish were then decapitated. The trunk samples were frozen in individual 1.5 ml tubes and stored at −20 °C for cortisol analysis.

Cortisol extraction and assay

The cortisol extraction and assay was done by slightly modifying previously published procedures (Cachat et al., 2010; Canavello et al., 2011). In brief, trunk samples were thawed and weighed, and subsequently homogenized with phosphate-buffered saline (PBS). Diethyl ether was added to the homogenates and centrifuged for 15 min at 4 °C. The ether layer containing cortisol was isolated in a separate tube. The addition of ether, centrifugation, and ether isolation was repeated for a total of three times, collecting all three ether layers in one tube for each sample. The ether was then dried under a light stream of air until only a yellow oil containing the cortisol remained in each tube. The oil in each tube was reconstituted with PBS and refrigerated overnight (4 °C). Cortisol was quantified via an enzyme-linked immunosorbent assay (ELISA) as per the manufacturer’s instructions (Salimetrics, State College, PA).

Data analysis

A priori sample size calculations were conducted using G*Power software (Faul et al., 2007) using the following parameters: d = 0.95, α = 0.05, power = 0.95. Effect size was based on the effects of chronic unpredictable stress and prazosin on anxiety measures in previously published studies (Marcon et al., 2016; Singh et al., 2013).

Upon selection from the home tank on the day of the experiment, each fish was given a sample number. Behavioral and cortisol analyses were conducted; sample numbers were then matched with the treatment(s) and analyzed by group. Data are presented as the means and standard errors of the mean (SEM) for each group. Raw data (see Data S1) was processed using JASP software (University of Amsterdam, Amsterdam, The Netherlands, https://jasp-stats.org/). For Experiment 1, overall (6-minute) behavioral variables, cortisol, and trunk weights were compared by independent sample t-tests (with stress condition as the independent variable) and one-minute bin data for behavioral variables were compared by repeated-measures ANOVA. For Experiment 2, overall (6-minute) behavioral variables, cortisol, and trunk weights were compared by two-way ANOVA analyses (with stress condition and drug treatment as the independent variables) and one-minute bin data for behavioral variables were compared by repeated-measures ANOVA. Tukey post-hoc analyses were conducted when appropriate and Greenhouse-Geisser sphericity correction was made if Mauchly’s test of sphericity indicated a violation of the sphericity assumption for the repeated-measures ANOVA tests. A significance level of p < 0.05 was used as the criterion for results to reach statistical significance.

Results

Experiment 1: Effects of seven days of chronic unpredictable stress (CUS) treatment on behavioral measures in the novel tank test (NTT), basal levels of cortisol, and body weights in adult zebrafish.

Motor activity in the novel tank test

A t-test for independent means indicated no significant effect of chronic unpredictable stress on either the total distance traveled (t(48) = 1.274, p = 0.209) or mean ambulatory speed (t(48) = 1.077, p = 0.287) for the entire 6 min of the novel tank test (Table 2). When the total distance data was broken down into six 60-second bins (Fig. 1A) and analyzed with a repeated-measures ANOVA, there was no effect of stress (F(1, 48) = 1.551, p = 0.219), a significant effect of time (F(3.877, 186.085) = 7.403, p < 0.001), but no interaction between stress and time (F(3.877, 186.085) = 0.452, p = 0.765). For the mean ambulatory speed (Fig. 1B), again, there was no effect of stress (F(1,48) = 0.934, p = 0.339), a significant effect of time (F(4.077, 195.709) = 16.657, p < 0.001), but no interaction between stress and time (F(4.077,195.709) = 1.006, p = 0.406). These results show that the fish appeared to habituate after introduction to the novel tank, as the total distance per minute and mean ambulatory speed gradually increased across the duration of the novel tank test, but there was no effect of treatment on these measures.

Table 2 Overall behavioral measures of zebrafish in the novel tank test (Experiment 1).

Exposure to 7 days of chronic unpredictable stress (CUS) decreased the time spent in the top and marginally decreased the number of times fish entered the top zone of the novel tank in adult zebrafish compared to unstressed (control) fish (N = 25 in each group).

	Control	7d CUS					
Variable	M	SD	M	SD	t	df	p	Cohen’s d	
Total distance moved (cm)	1082.44	225.84	1191.60	364.03	1.274	48	0.209	0.360	
Mean ambulatory speed (cm/s)	5.03	0.81	5.30	0.96	1.077	48	0.287	0.305	
Time immobile (s)	13.26	9.99	15.42	17.11	0.544	48	0.589	0.154	
Number of entries to top	20.00	13.12	14.12	7.90	−1.920	48	0.061	−0.543	
Total time in top (s)	65.96	44.75	43.82	26.49	−2.129	48	0.038	−0.602	
Distance in top (cm)	233.58	160.16	182.32	112.61	−1.309	48	0.197	−0.370	

Figure 1 Motor measures of zebrafish in the novel tank test (Experiment 1).

There was a significant effect of time on (A) the total distance traveled per minute and (B) the mean ambulatory speed in the novel tank test (6 minutes). All fish generally swam longer distances at faster speeds due to habituation to the novel tank, but there was no effect of 7 days of chronic unpredictable stress and no interaction between stress and time on motor measures (N = 25 in each group).

Freezing behavior in the novel tank test

A t-test for independent means indicated no significant effect of chronic unpredictable stress on the total immobility time (t(48) = 0.544, p = 0.589) in the novel tank test (Table 2). When the immobility data was broken down into six 60-second bins (Fig. 2) and analyzed with a repeated-measures ANOVA, there was no effect of stress (F(1, 48) = 0.359, p = 0.552), no effect of time (F(2.191, 105.178) = 1.359, p = 0.262), and no interaction between stress and time (F(2.191, 105.178) = 1.114, p = 0.336). Thus, neither time nor treatment significantly altered freezing behavior across the six minutes of the novel tank test.

Figure 2 Freezing behavior of zebrafish in the novel tank test (Experiment 1).

There was no effect of treatment or time on the amount of time zebrafish spent immobile in the novel tank test (6 minutes, N = 25 in each group).

Exploratory behavior in the novel tank test

A t-test for independent means indicated a marginally significant effect of chronic unpredictable stress for the number of entries to the top zone (t(48) =  − 1.920, p = 0.061; CUS < untreated), a significant effect on the total time spent in the top zone (t(48) =  − 2.129, p = 0.038; CUS < untreated), but no significant difference in the distance traveled in the top zone (t(48) =  − 1.309, p = 0.197; Table 2). When the number of entries to the top zone was broken down into six 60-second bins (Fig. 3A) and analyzed with a repeated-measures ANOVA, there was a marginal effect of stress (F(1, 48) = 3.454, p = 0.069), a significant effect of time (F(3.855, 185.031) = 6.701, p < 0.001), but no interaction between stress and time (F(3.855, 185.031) = 0.935, p = 0.442). For the time spent in the top zone (Fig. 3B), there was a significant effect of stress (F(1, 48) = 4.530, p = 0.038), a significant effect of time (F(3.540, 169.936) = 13.317, p < 0.001), but no interaction between stress and time (F(3.540, 169.936) = 1.318, p = 0.268). For the distance traveled in the top zone (Fig. 3C), there was no effect of stress (F(1, 48) = 1.635, p = 0.207), a significant effect of time (F(3.631, 174.291) = 12.430, p < 0.001), but no interaction between stress and time (F(3.631, 174.291) = 1.408, p = 0.237). Similar to the motor measures, the fish appeared to habituate and explore more of the top zone of the novel tank over time, but the fish that were chronically stressed generally explored the top zone less than the fish that were left untreated for seven days.

Figure 3 Exploratory measures of zebrafish in the novel tank test (Experiment 1).

There was a significant effect of time on all top measures in the novel tank test. Additionally, 7 days of chronic unpredictable stress (CUS) (A) marginally decreased the number of times zebrafish entered the top zone, (B) significantly decreased the amount of time zebrafish explored the top zone, and (C) slightly (but non-significantly) decreased the distance traveled in the top of the novel tank test over 6 minutes. Thus, all zebrafish tended to explore the top zone more across the duration of the test; however, chronically stressed fish explored less than non-stressed fish (N = 25 in each group).

Trunk cortisol

A t-test for independent means indicated that fish exposed to seven days of chronic unpredictable stress (CUS) had decreased basal levels of trunk cortisol compared to untreated control fish (t(48) =  − 3.130, p = 0.003; Fig. 4A). This finding suggests that seven days of chronic unpredictable stress decreases basal levels of cortisol in zebrafish compared to fish that were untreated.

Figure 4 Cortisol and body weight measures of zebrafish (Experiment 1).

Seven days of chronic unpredictable stress (CUS) significantly decreased (**p < 0.01) basal levels of trunk cortisol (A) but did not alter the body weight (B) of zebrafish compared to fish that were left untreated (N = 25 in each group).

Trunk weights

A t-test for independent means indicated that fish exposed to 7 days of chronic unpredictable stress (CUS) had similar trunk weights as fish that were left untreated for 7 days (t(48) =  − 1.111, p = 0.272; Fig. 4B). This finding suggests that 7 days of chronic unpredictable stress does not appear to alter factors involved with body weight regulation, such as feeding, in zebrafish.

Experiment 2: Effects of seven days of chronic unpredictable stress (CUS) treatment and seven days of prazosin treatment on behavioral measures in the novel tank test (NTT), basal levels of cortisol, and body weights in adult zebrafish.

Motor activity in the novel tank test

A two-way ANOVA indicated that there was no effect of stress (F(1,65) = 0.358, p = 0.552), no effect of drug treatment (F(1,65) = 0.139, p = 0.710), and no interaction between stress and drug (F(1,65) = 0.103, p = 0.750) on the total distance traveled in the novel tank test (Table 3). A two-way ANOVA indicated that there was no effect of stress (F(1,65) = 0.657, p = 0.421), no effect of drug treatment (F(1,65) = 1.186, p = 0.280), and no interaction between stress and drug (F(1,65) = 0.355, p = 0.553) on the mean ambulatory speed of fish in the novel tank test (Table 3). The total distance (Fig. 5A) and mean ambulatory speed (Fig. 5B) data was also broken down into six 60-s bins and analyzed with a repeated-measures ANOVA (see Table 4 for statistical analyses). These results indicate that, similar to the results from Experiment 1, fish appear to habituate to the novel tank across the duration of the test, but that there is no effect of stress or drug treatment on these measures of motor activity.

Table 3 Overall behavioral measures of zebrafish in the novel tank test (Experiment 2).

In adult zebrafish, exposure to 7 days of chronic unpredictable stress (CUS) increased the distance traveled in the top and the number of top zone entries whereas 7 days of chronic prazosin treatment increased the time spent in the top of zone of the novel tank test (N = 16 − 18 in each group). See text for results of significance testing.

	Control/Vehicle N = 16	Control/Prazosin N = 18	CUS/Vehicle N = 17	CUS/Prazosin N = 18	
Variable	M	SD	M	SD	M	SD	M	SD	
Total distance moved (cm)	1075.90	611.65	1070.93	206.33	1162.93	349.64	1097.22	320.95	
Mean ambulatory speed (cm/s)	5.83	2.60	5.21	1.07	5.31	1.05	5.13	0.96	
Time immobile (s)	67.71	108.90	24.05	28.06	24.06	27.47	19.93	18.79	
Number of entries to top	9.19	9.68	17.39	10.49	19.47	13.07	20.94	13.30	
Total time in top (s)	34.78	40.07	67.42	44.41	63.59	45.87	74.91	49.61	
Distance in top (cm)	140.69	160.39	236.92	173.72	294.23	207.33	271.54	161.64	

Figure 5 Motor measures of zebrafish in the novel tank test (Experiment 2).

In general, zebrafish gradually increased the total distance traveled per minute (A) and demonstrated increased ambulatory speeds (B) across the duration of the novel tank test (6 minutes), but there was no effect of chronic stress or drug treatment on these measures (N = 16 − 18 in each group).

Table 4 Results of repeated measures ANOVA (Experiment 2).

Adult zebrafish from all treatment groups generally habituated to the novel tank across the duration of the test (6 minutes), but there was no significant interaction between time and drug treatment or stress on any of the behavioral measures of anxiety (N = 16 − 18 in each group). Significance (p < 0.05, indicated with bold text) for all dependent variables were determined with a repeated-measures ANOVA with Greenhouse-Geisser correction.

	Total distance (cm)	Mean ambulatory speed (cm/s)	Time immobile (s)	Number of entries to top	Total time in top (s)	Distance in top (cm)	
	F	p	F	p	F	p	F	p	F	p	F	p	
Within-subjects Effects	
Time	9.795	<0.001	2.022	0.133	15.136	<0.001	7.112	<0.001	15.075	<0.001	13.679	<0.001	
Time* Stress	4.262	0.005	2.134	0.118	1.705	0.190	0.766	0.545	0.375	0.822	0.898	0.452	
Time* Drug	0.573	0.644	0.240	0.802	0.948	0.381	0.425	0.785	0.215	0.927	0.350	0.811	
Time* Stress* Drug	1.811	0.136	3.226	0.039	0.392	0.651	0.771	0.541	0.920	0.451	0.620	0.620	
Between-subjects Effects	
Stress	0.308	0.581	0.118	0.732	3.050	0.085	5.886	0.018	2.768	0.101	5.100	0.027	
Drug	0.191	0.663	0.050	0.824	2.931	0.092	3.131	0.082	4.062	0.048	0.750	0.390	
Stress* Drug	0.100	0.753	0.123	0.727	2.018	0.160	1.431	0.236	0.956	0.332	1.876	0.176	

Freezing behavior in the novel tank test

A two-way ANOVA indicated that there was a marginal effect of stress (F(1, 65) = 3.050, p = 0.085; CUS < control), a marginal effect of drug treatment (F(1, 65) = 3.050, p = 0.085; prazosin < vehicle), but no interaction between stress and drug (F(1, 65) = 2.087, p = 0.153) on total immobility time in the novel tank test (Table 3). The immobility data (Fig. 6) was also broken down into six 60-second bins and analyzed with a repeated-measures ANOVA (see Table 4 for statistical analyses). These results indicate that immobility generally decreases across the duration of the novel tank test, but that there is no significant effect of stress or drug treatment on this behavioral measure.

Figure 6 Freezing behavior of zebrafish in the novel tank test (Experiment 2).

In general, zebrafish spent less time immobile across the novel tank test (6 minutes), but there was no significant effect of chronic stress or drug treatment on immobility (N = 16 − 18 in each group).

Exploratory behavior in the novel tank test

A two-way ANOVA indicated that there was a significant effect of stress (F(1, 65) = 5.939, p = 0.018; CUS > control), a marginally significant effect of drug treatment (F(1, 65) = 2.903, p = 0.093; prazosin > vehicle), but no interaction between stress and drug (F(1, 65) = 1.404, p = 0.240) on the number of entries to the top zone (Table 3). A two-way ANOVA indicated that there was no effect of stress (F(1, 65) = 2.769, p = 0.101), a significant effect of drug treatment (F(1, 65) = 4.061, p = 0.048; prazosin > vehicle), but no interaction between stress and drug (F(1, 65) = 0.956, p = 0.332) on the total amount of time spent in the novel tank test (Table 3). A two-way ANOVA indicated that there was a significant effect of stress (F(1, 65) = 4.876, p = 0.031; CUS > control), no effect of drug treatment (F(1, 65) = 0.745, p = 0.391), but no interaction between stress and drug (F(1, 65) = 1.947, p = 0.168) on the distance traveled in the top zone of the novel tank (Table 3). The number of entries to the top (Fig. 7A), time spent in the top zone (Fig. 7B), and distance traveled in the top zone (Fig. 7C) were also broken down into six 60-second bins and analyzed with a repeated-measures ANOVA (see Table 4 for statistical analyses). These results suggest that allowing additional time (seven days) to elapse between the chronic stress paradigm and testing in the novel tank perhaps reverses the deficits in top zone exploration elicited by CUS observed in Experiment 1. In addition, these results suggest that chronic prazosin increases exploration in the novel tank test in the absence of stress, but does not appear to alter any stress-induced effects on top zone exploration in the novel tank when prazosin treatment follows chronic unpredictable stress.

Figure 7 Exploratory measures of zebrafish in the novel tank test (Experiment 2).

In general, zebrafish explored the top of the novel tank more across the duration of the novel tank test (6 minutes) by (A) entering the top zone more frequently, (B) spending more time in the top zone, and (C) traveling a longer distance in the top zone. Zebrafish exposed to chronic unpredictable stress (CUS) for 7 days before drug or vehicle treatments entered the top zone significantly more times and traveled a significantly longer distance in the top compared to non-stressed subjects. If fish were chronically treated with prazosin, they spent significantly more time in the top of the tank compared to vehicle-treated fish (N = 16 − 18 in each group).

Trunk cortisol

A two-way ANOVA indicated that there was no effect of stress (F(1, 65) = 2.511, p = 0.118), no effect of drug treatment (F(1, 65) = 0.624, p = 0.432), and no interaction between stress and drug (F(1, 65) = 0.636, p = 0.428) on basal levels of cortisol (Fig. 8A). Although these results did not reach statistical significance, the fish subjected to chronic unpredictable stress and subsequently were vehicle-treated had lower levels of cortisol than untreated-vehicle controls, which is similar to the pattern of results observed in Experiment 1. Prazosin treated-fish also had lower levels of basal cortisol compared to unstressed/vehicle-treated fish; however, this finding did not reach statistical significance.

Figure 8 Cortisol and body weight measures of zebrafish (Experiment 2).

Neither chronic unpredictable stress treatment (CUS) and nor drug treatment significantly altered basal levels of trunk cortisol (A) or body weight (B) of zebrafish (N = 16 − 18 in each group).

Trunk weights

A two-way ANOVA indicated that there was no effect of stress (F(1, 65) = 0.057, p = 0.811), no effect of drug treatment (F(1, 65) = 0.173, p = 0.679), and no interaction between stress and drug (F(1, 65) = 0.467, p = 0.497) on subject trunk weights (Fig. 8B). Similar to the results from Experiment 1, these results indicate that neither stress treatment nor chronic drug treatment altered factors involved with body weight regulation.

Discussion

The noradrenergic system is critical for the regulation of several functions, including the regulation of stress responses. The locus coeruleus, the major noradrenergic nucleus of the brain, supplies norepinephrine both systemically and directly to regions throughout the brain including the amygdala, hypothalamus, and the medial prefrontal cortex (mPFC), all areas involved with regulating responses to stress. Noradrenergic dysfunction has been hypothesized to be involved with the neuropathology associated with PTSD (Hendrickson & Raskind, 2016; O’Donnell, Hegadoren & Coupland, 2004; Southwick et al., 1999a; Southwick et al., 1999b; Strawn & Geracioti, 2008). Dysregulation of the noradrenergic system may ultimately contribute to the alterations in the function of the physiological stress axis observed in patients and animal models. Thus, pharmacological agents that target the norepinephrine regulation of stress responses have the potential to normalize neuroendocrine dysfunction associated with stress-related conditions.

The hypothesis of the current study was that the zebrafish model of CUS would increase basal levels of cortisol and increase anxiety-like behavior in the novel tank test, and that chronic prazosin treatment would reverse alterations induced by chronic stress. In support of this hypothesis, CUS slightly increased anxiety-like behavior in the novel tank test; however, this change in behavior was associated with significantly lower levels of basal cortisol. In addition, prazosin appeared to decrease levels of anxiety-like behavior in the absence of CUS. However, it also appeared that CUS-induced decreases in exploratory behavior and basal cortisol levels started to normalize (and in the case of the behavioral variables, reversed) in the 7d interim when vehicle and drug treatment were being administered between CUS and dependent variable assessments.

The study design and timing of the assessment of the dependent variables should be considered when interpreting the results of the current study. In the first experiment, behavioral and neuroendocrine measurements were assessed immediately after the seven days of chronic stress. In the second experiment, subjects were exposed to seven days of chronic stress followed by a week of chronic drug treatment administered in the absence of unpredictable stressors. It is possible that any stress-induced effects may have been blunted or reversed by the time the measurements were assessed. For example, in Experiment 1, fish that were exposed to CUS had lower levels of basal cortisol compared to unstressed fish. In Experiment 2, fish that were chronically stressed but subsequently treated with vehicle for seven days still had lower basal cortisol levels than non-stressed, vehicle treated subjects; however, this difference did not reach the criterion for statistical significance (see Fig. 8A). Alternatively, in Experiment 2, the additional week of handling necessary to administer the drug/vehicle treatment may have triggered adaptive mechanisms in animals previously exposed to the CUS paradigm, which could have increased the elevated exploratory behavior observed in the novel tank test compared to non-stressed, vehicle-treated controls (see Fig. 7). Baseline measures of anxiety could have been increased with the 7 additional days of handling; this supposition is supported by the fact that unstressed/vehicle-treated fish in Experiment 2 generally displayed more anxiety-like behavior (increased immobility and decreased top zone exploration) compared to the unstressed fish from Experiment 1 (see group means in Tables 2 and 3). Future studies should address the duration or persistence of long-term neuroendocrine and behavioral effects of 7d CUS and whether chronic prazosin treatment administered during the same period as the stressors would circumvent any possible effects of allowing previously stressed animals to adapt back to non-stressed conditions. Other studies have examined the effects of anxiolytic compounds administered concurrent with chronic stress paradigms; for example, in a previous study, zebrafish were exposed to chronic unpredictable stress for five weeks but then were treated with the antidepressant fluoxetine during the last 8 days of the stressor paradigm (Song et al., 2018). Another study exposed zebrafish to chronic unpredictable stress for 14 days but treated with the putative anxiolytic N-acetylcysteine concurrent with the last 7 days of the stress (Mocelin et al., 2019). Thus, it would be interesting to see whether continuing the chronic unpredictable stress exposure during treatment or by conducting the stress exposure during the dark phase (Manuel et al., 2014) would make prazosin-induced effects more obvious.

In the current report, prazosin treatment alone appeared to enhance exploratory behavior in the novel tank test but did not affect CUS-induced changes in anxiety-like behavior. In addition, prazosin-treated groups had lower levels of cortisol than non-stressed, vehicle-treated subjects, although this decrease did not reach statistical significance. Prazosin treatment may function to generally prevent reactions to stress (Rasmussen, Kincaid & Froehlich, 2017); thus, further studies should also determine whether preventative prazosin treatment is effective at blocking any chronic or acute stress effects on anxiety-like behavior or neuroendocrine dysfunction. These studies would help clarify whether prazosin would be clinically efficacious, not by normalizing the effects of trauma or stress, but by preventing any responses to further triggering stimuli that may elicit PTSD symptomology. As prazosin acts by putatively blocking α1 receptors, antagonism of alpha-1 receptors could possibly prevent further stress or trauma from triggering the norepinephrine-mediated stimulation of stress axis reactivity (Ma & Morilak, 2005). Thus, prazosin may work better as a prophylactic in treating stress-related conditions, as has been observed for other medications (Roque, 2015), although much more clinical work would be necessary to establish this as a potential option for therapy.

It is also interesting to note that, although it was expected that seven days of CUS treatment would elicit increases in basal cortisol levels based on previously published reports (Manuel et al., 2014; Marcon et al., 2016; Marcon et al., 2018; Piato et al., 2011), the results from the first experiment indicate that the CUS protocol can elicit hypocortisolic responses. There are several factors that could explain the different results between laboratories, such as the source, strain, previous stress exposure, or age of the subjects. For example, one previous report indicated that there are possible sex-specific differences in basal cortisol levels after exposure to chronic stress, with male zebrafish exhibiting increases in cortisol compared to untreated controls, while the levels of basal cortisol did not change in females relative to untreated controls (Rambo et al., 2017). Thus, the impact of both prazosin and chronic stress on our dependent measures may be masked by including both sexes in the analyses. The current results support a recent call for much more research into the housing, breeding, and other husbandry conditions that may be contributing factors to differences in experimental results between laboratories (Lidster et al., 2017; Tsang et al., 2017; Varga, Ekker & Lawrence, 2018).

Another factor that could play a role in our findings is the binding profile of prazosin to α1 adrenergic receptors in the zebrafish brain. Although the binding of prazosin in brain has been extensively studied in rodent and other mammalian models (Dashwood, 1982; Greengrass & Bremner, 1979; Mignot et al., 1989; Morrow et al., 1985; Morrow & Creese, 1986; Rainbow & Biegon, 1983), the same cannot be said about prazosin binding in zebrafish brain. Studies using preparations of codfish brain indicate that prazosin binding may in fact be different in fish brains compared to rodent brains (Bergström & Wikberg, 1986a; Bergström & Wikberg, 1986b). Perhaps more importantly, although zebrafish α2 and β receptor binding and distribution has been explored (Ampatzis & Dermon, 2010; Ampatzis, Kentouri & Dermon, 2008; Ruuskanen et al., 2005a; Ruuskanen et al., 2005b; Wang et al., 2009), little comparable information on zebrafish α1 receptor binding and distribution is available in the extant literature. Thus, more work is necessary for a complete understanding of noradrenergic modulation of stress responses in the zebrafish model.

In sum, this study suggests that the clinical efficacy of prazosin in reducing symptoms of stress-related conditions like PTSD does not involve the normalization of physiological stress axis dysfunction. Rather, the clinical effectiveness of prazosin likely involves other mechanisms of altering stress regulation. In addition, this study also highlights the importance of considering methodological and husbandry factors when interpreting results across several vertebrate animal studies, which will ultimately contribute to a better understanding of the complex nature of the regulation and expression of stress responses.

Conclusions

This study demonstrated that seven days of chronic unpredictable stress exposure in zebrafish increased the expression of anxiety-like behavior and decreased basal levels of cortisol. When prazosin, a putative alpha-1 receptor antagonist, was chronically administered to subjects after the stress exposure, stress-related effects on behavior and hormones were not reversed; however, prazosin appeared to decrease anxiety-like behaviors in the novel tank test in the absence of stress exposure. Further studies are necessary to determine the longevity of chronic stress-induced responses and whether effects on stress-induced alterations in stress responses are dependent on the timing of drug treatment. These studies also suggest that normalization of neuroendocrine dysfunction may not be involved with the clinical efficacy of prazosin in the treatment of PTSD, although human studies are needed to confirm this finding.

Supplemental Information

Data S1 Raw data for Experiment 1 and Experiment 2: behavioral measures and whole-body cortisol levels in adult zebrafish

The raw data from the behavioral and cortisol analyses from Experiment 1 (overall and one minute bin data) and from Experiment 2 (overall and one minute bin data).

Click here for additional data file.

The authors would like to thank Dr. Harold Olivey and Dr. Jenny Fisher of the Indiana University Northwest Department of Biology for their technical assistance with some aspects of these studies.

Additional Information and Declarations

Competing Interests

Author Contributions

Data Availability

The authors declare there are no competing interests.

Michael P O’Daniel conceived and designed the experiments, performed the experiments, analyzed the data, authored or reviewed drafts of the paper, and approved the final draft.

Maureen L Petrunich-Rutherford conceived and designed the experiments, performed the experiments, analyzed the data, prepared figures and/or tables, authored or reviewed drafts of the paper, and approved the final draft.

The following information was supplied regarding data availability:

The raw data is available in the Supplemental File.

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
