# Peer review of "Effects of chronic prazosin, an alpha-1 adrenergic antagonist, on anxiety-like behavior and cortisol levels in a chronic unpredictable stress model in zebrafish (Danio rerio)"

_PeerJ, doi:10.7717/peerj.8472_

## Round 0.1 · original submission · Major Revisions

The zebra fish experts have raised a significant number of points about the design and interpretation of your experiments that all require addressing in any revised manuscript. My only additional comment at this point is around the receptor binding profile of prazosin at zebra fish receptors - what evidence is there that prazosin will be exerting its effects at the same type of receptor in fish as in humans ? Has the pharmacological profile of zebra fish alpha1 receptors been investigated?

Reviewer 1 ·

Basic reporting

This manuscript entitled “Effects of chronic prazosin, an alpha-1 adrenergic antagonist, on anxiety-like behavior and cortisol levels in a chronic unpredictable stress model in zebrafish (Danio rerio)” studied the effects of prazosin on behavioral and biochemical parameters in adult zebrafish submitted to unpredictable chronic stress protocol. The manuscript presents appropriate methodological aspects, the text is clear and unambiguous, and the article presents a professional structure. The statistical analyses are correct and properly reported. Figures, table and raw data are well presented.

Experimental design

The description of the materials and methods has some limitations. Important aspects regarding animal care and randomization were not adequately described. I suggest that the authors review the ARRIVE and PREPARE guidelines to better describe the methodology. What was the method used for random allocation in the treatment groups? Were experimenters blind to treatment? Were data analysts blind to treatment? These questions need to be addressed and clearly stated in the methods section.

I believe that beyond temperature, physical-chemical characteristic of water (for example, pH, conductivity) should be controlled during the experiments. It is not described in the methods section.

Did the authors consider the potential effect of the tank on statistical analysis?

Were experimenters blinded throughout the experimental protocols? If not, there is a possibility of bias in the interpretation of the data. This information should be described in the M&M section.

How was the sample size calculated? Was it based on a pilot experiment to determine the effects sizes? How were the calculations performed?

Did the authors check for sex effect in these tests? It is a relevant question because female and male zebrafish can respond in a different manner. In my opinion, is mandatory to check for sex effect. If there are no effects of the sex, then the data can be pooled.

How many times the prazosin solution was used?

Line 185, please change dose to concentration.

There is a clear difference between immobility time in control groups (please see figure 2 and 6; table 2 and 3). How could the authors explain this difference?

Please describe how much food was used. Was it 1% body weight per day? Please give a figure.

Validity of the findings

Although I think this approach interesting, I suggest another experimental design for the next experiments. In Marcon et al. (2019), the animals were submitted during 14 days to UCS. In the last 7 days, zebrafish were treatment and stressed concomitantly. I believe that this protocol will be more suitable to demonstrate the potential effect of the prazosin as a treatment of stress-related disorders.

·

Basic reporting

The writing is professional and intelligible, but there are some typos, some areas lack clarity, are repetitive, and/or require more work. Literature references and overall structure are good. There are some improvements needed, for example (not comprehensive list, please review the whole manuscript more carefully):

- use of word 'alter' instead of clear direction of effect (increase/decrease) throughout the manuscript, particularly problematic in the abstract and figure/table captions.
- Line 61, 'resilience of treatment' should read 'resilience to treatment'
- Line 81-84, the claim that prazosin blocks noradrenergic regulation of the HPA axis via alpha-1 receptor is highly problematic here. The authors do not cite any existing literature to support this claim and the experimental design of this study is not suited to answer whether prazosin blocks noradrenergic receptors.
- Line 98-101, this sentence implies that CUS models aspects of MDD and PTSD. Yet, the authors list two review papers (Caramillo et al., 2015; Stewart et al., 2014) that both 'suggest' that CUS (among many others) may be used to model stress/PTSD but do not show contain experimental work. In general, it is not obvious why 'chronic' mild stress would be a good model for PTSD, even when PTSD may result from a single traumatic experience. It may be more appropriate to argue that the current study is modeling general anxiety and/or stress that is shared between many mood disorders, anxiety disorders and trauma and stressor-related disorders, instead of only use the PTSD framework.
- Line 203, typo in euthanasia
- Line 313-314, and 326 please confirm that the statistics are correct here

Experimental design

The mechanisms underlying PTSD-related HPA-axis activation and behind prazosin's effectiveness are worthy topics. The research question is relevant and meaningful. It is also well-defined in the manuscript but lack key details about operationalization and procedures. Thus, it is not clear which of the measurements are relevant for PTSD modeling and why. For example, the authors measure locomotor activity but do not state whether this is to determine stress-related responses of hyperactivity/hypoactivity.

Other general and specific comments:

- Given that ample literature exists on the importance of prior housing, breeding practices and husbandry conditions for behavioural studies (some of which are mentioned in line 416-419), it is not clear why the authors used fish that were bought from a supplier and not bred in-house, a practice that would allow more control and consistency over the heterogeneity that may exist in wild-type fish. Similarly, is it possible that short acclimatisation of a week and high stocking density (5-7 fish in 1.8L tanks) may have caused stress that may have led to lack of significant differences between control and 'stressed' groups?

- Line 131 & 142/152. The information about housing densities does not add up. If 5-7 fish were places in 1.8L tanks (line 131), how were 50 fish were put in four tanks (line 142, implying 12 fish per tank) and 72 fish were allocated into eight tanks (line 152, implying 9 fish per tank). Similarly, in line 177, for crowding, how many fish were put in the 200 ml water?
- Line 133, please include information on any/all other parameters of housing water quality: pH, salinity, oxygenation, ammonia, nitrates/nitrite, etc.
- Line 133-134, when were fish fed on experimental days or drug exposure days? Were they fed before or after the CUS/prazosin treatment?
- Line 147 and 160, what does subsequently mean here? How long after the novel tank test were fish euthanized?
- Line 152-157, does 'handled/unhandled' mean stressed/unstressed or something else here?
- Line 153, how long was the acclimation period for experiment 2?
- Line 155-160, please stated explicitly whether the fish received stress and prazosin treatment concomitantly or in sequence. Line 158-160 seems to be the same sentence as line 145-147 and implies that fish were euthanized after the CUS treatment.
- Line 172-174 (a), while drastic temperature changes can be considered stressful, given that the fish were kept at 26 +/- 2C, is 23C and 33C drastic enough when housing condition ranges of 24-28C? Besides existing literature (Piato et al, 2011, Marcon et al, 2016, and Rambo et al, 2017) that does the same without much explanation, why do the authors consider 23C and 33C as stressful for a fish species that can tolerate a wide temperature range in the wild and in captivity? Is this ethologically relevant?
- Line 172-174 (b), how did the experimenters maintain the temperatures at 23C and 33C for 30 min? Did the heaters/coolers used for this make noise or vibrations? Is it possible that a lack of water circulation (and re-oxygenation) for 30 minutes is perhaps more of a stressor than the temperature changes?
- Line 181-191, please provide an explanation for placement of individual fish during prazosin exposure in 50 mL of water, esp given that social isolation and little water were used as stressors (line 175 and 178).
- Line 193-204, what was temperature of the testing tank? pH? Salinity? Lighting conditions? Did the authors use system water to fill the testing tank? How often was the water changed between different trials?
- Line 194-196, please provide an explanation for why fish were acclimatized to the experiment room prior to a test of response to 'novelty'? Were the home tanks aerated during these 30 minutes? Were all fish in a home-tank tested at once or in sequence such that some fish were acclimatized for longer time?
- Line 197, please provide an explanation for using a tank of same size and dimensions as the home tank for a test of response to 'novelty'?
- Line 199, please cite previously existing literature that has validated or used this software for tracking zebrafish.
- Line 200-203, please provide more analytical information for this section. Why are distance and speed (usually quantifying hyperactivity) not considered measures of anxiety but only of general motor activity? What does 'top of the tank' means? Does the 'top' refer to top-half, top 25%, or something else? How long was the water column/height? Also, why did the authors report on 'top' and 'immobility' behavior and not include other relatively easily quantifiable measures of anxiety, bottom-dwelling, erratic movement, latency to enter 'top', and pigmentation change as measures of anxiety?
- Line 203, typo in euthanasia
- Line 207-209, please give an estimate for how long did euthanasia take typically? 1 min? 10 min? 30 min?
- Line 213, were the trunks were cleaned of digestive tract and eggs? If not, please indicate in methods the timing of last feeding prior to euthanasia and if it was the controlled for all fish.

Validity of the findings

Given the findings of experiment 1, that CUS paradigm used here had statistically significant effect on only one measure (total time spent in the top zone) out of six behavioural, cortisol, and body weights, why did the authors choose to employ the exact same CUS protocol in experiment 2?
- Line 283, given that there was an effect on time spent in the top zone, but no effect on distance traveled, and only a marginal effect of entries, it is justified to say that 'chronically stressed generally explored the top zone less'?
- Line 313-314, please confirm that the statistics for both stress (F(1,65) = 3.050, p = 0.085) and drug (F(1,65) = 3.050, p = 0.085) are exactly the same and this is not a typo or a copy/paste error.
- Line 318, please rephrase 'no strong effect' to better represent the lack of statistically significant differences. Also 'across the novel tank test' should be clarified to better represent the temporal nature (and not spatial) of these results.
- Line 326, please confirm that the F-value is correct here (F(1,65) = 956, p = 0.332).
- Line 333-335, given this statement, is it use of CUS and NTT as employed in this study appropriate for modeling persistent/long-occurring PTSD or even long-term effects of stress?
- Line 345-347, given this, is it possible that the procedure for vehicle/prazosin exposure (and/or the housing conditions) may be more stressful than the CUS itself?
- Line 374-376, this statement is misleading, as it implies that there were 'any effects of CUS' on behavior and hormone, which prazosin did not mitigate. Please rephrase this sentence to more accurately represent your data, control animals had more stress than CUS so it is not possible to determine the prazosin effects.
- Line 384-400, given previously existing literature (Song et al.,2018 Rasmussen et al., 2017, and similar studies supporting simultaneous stress and treatment), and Piato et al., 2011 (showing optimal stress after 7 days but not at 14 days) please elaborate on the choice of sequential timeline (CUS followed by prazosin) and stress responses collected on day 14 in experiment 2 (instead of day 7). This is particularly relevant for modeling of PTSD, as experiment 2 results (non-prazosin groups) do not replicate the CUS-effect findings from experiment 1, and are actually in the opposite direction.
- Line 405-406, this conclusion is problematic. The authors do not cite any existing literature to support this claim and the experimental design of this study does not substantiate that prazosin prevents further triggering of PTSD.
- Line 413, did the authors analyze the effect of sex? Especially given the relatively good sample size in experiment 1, it may be possible to see if there is an interaction between sex and CUS for cortisol?
- Line 420-423, again, this summary should be rephrased to more accurately represent the data. Same applies to the 'Conclusion' section.
- Line 430, please state the direction of the result, instead of 'alter'.
- Figures 1-8, why is minute 7 included in the figures when the testing only lasted 6 minutes?
- Figure 1 and 3. There are symbols denoting the effect of time which make these more figures complicated. Is this necessary or relevant? On the other hand, none of the significant effects are highlighted in the figures or the captions. Why?
- Figure 3 and 7, the captions are misleading. Please state the specific lack/presence of statistically significant effect of CUS and/or prazosin clearly in the caption and indicate it on the graphs. It may be of benefit to represent the data in graphs C (for both 3 and 7) as the % of total distance traveled that is in the top.
- Figure 1-8, it would help the reader if you keep the legends (colours and shapes) for the non-drug groups consistent across figures for the two experiments. In figures 1-3, red-filled squares are used for CUS that are not treated with prazosin, but the comparable groups from experiment 2, are unfilled-squares (and red-squares are CUS with prazosin).
Table 1, caption, 'throughout the light period between 9 a.m. and 4 p.m.' Please rephrase this as 9am-4pm covers only half of the 14-hrs of light.
Table 2- captions are misleading when stating 'CUS altered some behavioral measures' when the table has only one or very few statistically significant effects listed.

Additional comments

The manuscript identifies an important research question about underlying mechanism behind prazosin use for PTSD. Both effort towards animal modeling of PTSD and use of zebrafish is commendable and are strength of the paper. The weaknesses are poor construct validity, lack of key methodological details, and exaggerated implications of results.

Other specific comments:
- There is no mention of what generally causes/leads to PTSD (traumatic events) in the introduction. Prevalence rates in the US are cited from 2007. More recent and global rates may help make the need for a treatment more obvious. Similarly, while it is noted that active duty service members are prescribed more benzodiazepines for PTSD, it is not obvious that mentioned population or other people experiencing war/conflict situations have higher risk/prevalence of PTSD.

---

## Round 0.2 · accepted · Accept

The authors have provided a thoughtful and comprehensive response to the comments of the Reviewers.

Reviewer 1 ·

Basic reporting

no comment

Experimental design

no comment

Validity of the findings

no comment

Additional comments

In the present form the manuscript can be accept. The authors responded properly all my issues.

·

Basic reporting

The authors have addressed my concerns and provided sufficient counter-argument or fixed the issue.

Experimental design

The authors have addressed my concerns and provided sufficient counter-argument or fixed the issue.

Validity of the findings

The authors have addressed my concerns and provided sufficient counter-argument or fixed the issue.